# Knowledge Development Trajectory of the Internet of Vehicles Domain Based on Main Path Analysis

**DOI:** 10.3390/s23136120

**Published:** 2023-07-03

**Authors:** Tang-Min Hsieh, Kai-Ying Chen

**Affiliations:** 1College of Management, National Taipei University of Technology, 1, Sec. 3, Zhongxiao E. Rd., Taipei 10608, Taiwan; 2Department of Industrial Engineering and Management, National Taipei University of Technology, 1, Sec. 3, Zhongxiao E. Rd., Taipei 10608, Taiwan; kychen@mail.ntut.edu.tw

**Keywords:** internet of vehicles, main path analysis, cluster analysis, sensor

## Abstract

The Internet of vehicles (IoV) is an Internet-of-things-based network in the area of transportation. It comprises sensors, network communication, automation control, and data processing and enables connectivity between vehicles and other objects. This study performed main path analysis (MPA) to investigate the trajectory of research regarding the IoV. Studies were extracted from the Web of Science database, and citation networks among these studies were generated. MPA revealed that research in this field has mainly covered media access control, vehicle-to-vehicle channels, device-to-device communications, layers, non-orthogonal multiple access, and sixth-generation communications. Cluster analysis and data mining revealed that the main research topics related to the IoV included wireless channels, communication protocols, vehicular ad hoc networks, security and privacy, resource allocation and optimization, autonomous cruise control, deep learning, and edge computing. By using data mining and statistical analysis, we identified emerging research topics related to the IoV, namely blockchains, deep learning, edge computing, cloud computing, vehicular dynamics, and fifth- and sixth-generation mobile communications. These topics are likely to help drive innovation and the further development of IoV technologies and contribute to smart transportation, smart cities, and other applications. On the basis of the present results, this paper offers several predictions regarding the future of research regarding the IoV.

## 1. Introduction

The Internet of vehicles (IoV) is an Internet of things (IoT)-based network in the area of transportation that involves sensors, network communication, data processing, and automation control. The IoV enables real-time information exchange among vehicles, drivers, pedestrians, and road infrastructure through vehicle-to-everything (V2X) communication and thus facilitates the convergence of mobile communication technology, information systems, and intelligent transportation [1]. Understanding of the applicability of the IoV differs among fields. Automobile manufacturers primarily consider the IoV to be a tool for the informatization and intelligentization of automobiles, providers of intelligent transportation system (ITS) solutions consider the IoV to be a tool for facilitating the intelligentization of transportation-related technology, and Internet companies consider the IoV to be a mobile information terminal that offers opportunities for business innovation. The Internet has become an increasingly prevalent feature in industrial operations. In addition, vehicles are likely to be more frequently used as open platforms for the integration of various domains, and the IoV is likely to be a core aspect of this process [2].

The IoV was first developed through vehicular ad hoc networks (VANETs). VANETs are used in vehicle-to-vehicle (V2V) and vehicle-to-infrastructure (V2I) communications to ensure road safety, facilitate navigation, and provide services. VANETs are a crucial component of ITSs. Current IoV networks include more forms of communication than do VANETs; for example, they include vehicle-to-pedestrian (V2P), vehicle-to-network, vehicle-to-grid (V2G), and vehicle-to-sensor (V2S) communication [3,4]. The IoV has led to ubiquitous V2X connectivity, with “X” potentially representing vehicles, roads, infrastructure, people, the cloud, or the Internet [5]. The emergence of fifth-generation (5G) and sixth-generation (6G) technologies, artificial intelligence, intelligent driving systems, big data, and cloud computing have enabled the IoV to transform vehicles into intelligent entities, and thus the IoV has strong industry potential [6].

Given the current data-driven age, the IoV was built using IoT infrastructure, equipment, and technology. The amount of information generated in the IoT network is continuously increasing, and this phenomenon has led to new challenges related to the efficient management and extraction of data in the network. Therefore, the reexamination and application of large-scale adjustments to the efficiency, complexity, interface, dynamics, robustness, and interactive aspects of the IoT are essential; the traditional IoT framework is no longer sufficient to meet current demands. The present study systematically analyzed the development of next-generation IoT technology [7]. Specifically, we created a systematic catalogue of the most recent developments in the field of IoT technology and presented a comprehensive overview of IoT technology with consideration of big data, data science, network science, and connection technology. In addition, we proposed a system for IoT classification based on the medium access control (MAC) and radio duty cycle layers of the IoT architecture. The MAC layer is responsible for coordinating and connecting IoT devices, and duty cycling is a fundamental process for wireless networks and an essential means of saving energy, which is crucial if nodes must operate continuously for several days at a time [8]. Furthermore, in the current study, we focused on the conceptualization, key concepts, growth, and most recent trends of the IoT and discussed the importance of integrating big data, data science, and network science within the IoT. Additionally, we conducted in-depth analyses of the challenges associated with IoT networks, such as the construction of IoT frameworks, verification of data sources, and integration of cloud computing and edge computing into IoT networks to broaden the scope of the IoT’s applications. These challenges must be addressed through technological advancements in IoT systems before the IoV can be further developed.

Data from a preliminary study published by multiple automotive sales tracking companies, including LMC Automotive and EV-Volumes.com, revealed that approximately 10.5 million battery electric vehicles and plug-in hybrid electric vehicles were sold in 2022, constituting a 55% increase from the previous year. The data published in that study indicated that, overall, exponential growth had occurred in the electric vehicle industry and that this growth spurred the rapid development of IoV applications. In the current study, we explored how the IoV can improve the safety and efficiency of vehicles, advance automobile technology, protect the environment, and improve consumer quality of life. This paper proposes that the IoV could be used to address the following core issues: First, the IoV could improve traffic safety by enabling the establishment of V2V communication, making it possible for vehicles to share information regarding their speeds and locations relative to other vehicles. Such information sharing could then provide automatic responses that, in turn, could prevent traffic accidents. Second, the IoV could enhance traffic efficiency by facilitating V2V coordination, which would help to prevent traffic jams, enhance road-use efficiency, and enable the optimization of intelligent traffic signal settings. Third, the IoV is a key component of autonomous driving technology. By connecting to the IoV network, autonomous vehicles can obtain data regarding traffic conditions to improve the safety and efficiency of their autonomous driving. In addition, the IoV could be used to optimize vehicle paths and speeds in order to reduce unnecessary energy consumption and carbon emissions and thus alleviate the effects of vehicle use on global warming and climate change. Finally, the development of the IoV drives the development of new, innovative services and commercial models, such as shared transportation, unmanned vehicle use, and intelligent logistics models. These models may lead to new opportunities to improve human quality of life. Therefore, research regarding the IoV is potentially valuable and is thus worthy of continued investment and pursuit.

The current study was conducted as follows: First, main path analysis (MPA) was conducted to map the knowledge structure of IoV research. Next, cluster analysis was performed to select seven main clusters, which were identified as emerging topics for further analysis based on studies regarding the IoV published between 1993 and 2023. Accordingly, this paper provides suggestions regarding emerging topics. Our findings regarding these emerging topics may offer new perspectives on and new possibilities for the future development of the IoV.

### 1.1. Definition of the IoV

Contreras et al. [9] indicated that the IoV involves the seamless integration of IoT technology into in-vehicle communication systems and the combination of VANETs and the IoT. Alam et al. [10] defined the IoV as an integrated application system combining ITSs and the IoT to enable intelligent transportation and to connect vehicles to infrastructure as a means of creating vehicle–road networks that can provide seamless intelligent transportation services. Jiacheng et al. [11] described the IoV as a system of smart vehicles equipped with advanced sensors, controllers, actuators, and other devices in which communication and networking technology is employed for environmental sensing, intelligent decision making, and control. Hartenstein et al. [12] defined the IoV as an open, integrated, manageable, and reliable system that involves coordination among humans, vehicles, other machines, and the environment; that study also indicated that in the IoV, advanced information communication and processing technology is used to identify, process, and transmit static and dynamic information among humans, vehicles, communication networks, and road infrastructural devices and that the IoV therefore enables connectivity among humans, vehicles, and the environment.

This paper defines the IoV as an IoT network of transportation that involves sensors, wireless communication technology, satellite positioning systems, big data processing, and cloud computing. The IoV can be used to achieve intelligent management because it processes static and dynamic data related to vehicles, drivers, pedestrians, power grids, and road infrastructure and transmits these data to backend platforms. These data are subsequently analyzed, with the results used to enhance vehicle safety, manage traffic, control vehicles, facilitate vehicle diagnostics, and help drivers.

### 1.2. Architecture of the IoV

No consensus has yet been reached in academia regarding the structure of the architecture of the IoV. Nanjie [13] indicated that the architecture of the IoV is similar to that of the IoT; that is, the IoV comprises sensing layers (client), network layers (connection), and application layers (cloud). Studies have proposed IoV architectures with between three and seven layers (Table 1). The IoV is used for V2V communication and in vehicle-to-terminal service networks. It constitutes a complex system involving dynamic movement in addition to interaction among humans, vehicles, and the environment. In addition, the IoV relies on software for computation, deep learning, and communication and thus requires a strong architecture. Li et al. [14] proposed an architecture comprising four layers—namely a physical layer, a network layer, a middleware layer, and an application layer—for the IoV. The physical layer enables communication between vehicles and infrastructure and comprises hardware, sensors, and communication equipment. The network layer involves communication protocols and the infrastructure of the network and enables data transmission between vehicles and infrastructural devices. The middleware layer involves services and protocols that ensure compatibility among processes such as data management, data processing, and data sharing. Finally, the application layer involves practical applications and services, such as traffic management, navigation, and remote vehicle control. Table 1 presents a summary of the literature regarding the potential architectures of the IoV [1,2,5,9,13,14,15,16,17,18,19,20,21,22,23,24,25,26,27,28,29,30].

### 1.3. Applications of the IoV

The development of wireless communication and information technologies has led to the development of numerous applications of the IoV across multiple fields [1,32,33,34,35,36,37,38]. This subsection presents examples of such applications.

Safe Driving and Traffic Control: The IoV has enabled V2V communication and coordination, which has improved the safety of driving. Information sharing through the IoV has led to safer, more efficient, and more intelligent driving. Collision prevention systems integrated with the IoV can sense risks and warn drivers. In addition, in the event of a collision, the system automatically sends the location information of the vehicles involved and other data to emergency response teams, leading to faster rescue times, which in turn can help save lives. Finally, during traffic jams or after accidents, the IoV can be used to obtain real-time road condition and traffic data and thus facilitate the implementation of traffic control measures.

Convenient Services: The IoV provides drivers and passengers with real-time traffic information, weather updates, parking information, and navigation services, all of which enable drivers to avoid congestion or heavy traffic due to accidents. In addition, the IoV enables a driver to remotely lock and unlock their vehicle’s doors, start and stop their vehicle’s engine, and adjust their vehicle’s air conditioning, all of which save them time and make driving more comfortable and enjoyable.

Diagnostic and Remote Telematics: The IoV provides vehicle owners and technicians with access to remote diagnostic services. Technicians can understand and provide troubleshooting assistance for problems from remote locations, and sensors and real-time data collection enables technicians to monitor vehicle performance and identify problems, which saves time and reduces maintenance and repair costs. Finally, through 5G and wireless technology, vehicle owners and fleet managers can monitor vehicle performance, plan routes, and monitor driving behaviors to ensure vehicle safety, optimize vehicle utilization, and reduce fuel consumption.

Insurance: The IoV can be used to collect information regarding a vehicle’s speed, braking performance, and acceleration; such information can be used to assess driver risk and determine insurance premiums. In addition, insurance companies can use such information when people file claims and when companies need to process these claims; specifically, real-time data regarding the accident for which a claim is being filed can be sent to a company for assessment. In this way, the IoV can help create a streamlined insurance experience.

Infotainment: The IoV enables the use of various services for drivers and passengers. For example, the IoV can enable passengers to access music and video streaming services and make video calls through a high-speed data connection. In addition, because it collects and analyzes users’ preferences and habits, the IoV can provide data that enable content and services to be tailored to individual users’ needs and that therefore offer users customized experiences and make driving more enjoyable.

Self-Driving: Self-driving technology has gradually matured in recent years, and artificial intelligence and deep learning are crucial components of such technology. Self-driving technology can prevent traffic accidents due to drivers’ mistakes because the large amounts of data generated during self-driving improve the capability of deep-learning-based self-driving. In addition, advancements in hardware have rendered deep learning algorithms better able to process data in real time, which has improved environmental sensing, decision-making, and control abilities. In summary, integration of the IoV into self-driving technology systems can provide drivers with information regarding the environment, other vehicles, speed, and driving routes in real time, which in turn can improve driving safety.

## 2. Materials and Methods

### 2.1. Data Source

This study analyzed data retrieved from the Web of Science (WOS) academic database. The WOS database was established in 1997 by US-based company Thomson Reuters and provides research and citation data across numerous disciplines, including the natural sciences, engineering, medicine, agriculture, the humanities, and the social sciences.

### 2.2. MPA

Path citation analysis is a key research tool for bibliometric study. Such analysis is used to explore the relationships between source works and cited works by analyzing the citation relationships among research articles, authors, and journals. Analysis of citation behaviors in the literature can assist researchers in understanding the associations among studies, the citation characteristics of each discipline and domain, and the academic background of each author [39]. Citation analysis mainly involves the quantification of literature and the use of statistics, algorithms, and comparisons to analyze the quantified literature and the citation patterns in the literature. The results of such analysis reveal developmental trends, characteristics of literature use, the relevance of studies, and possible future trends within a domain [40,41]. We employed citation analysis as our main research approach and combined it with MPA and cluster analysis. First, citation analysis was performed to investigate the development in the academic knowledge regarding the IoV and the developmental trends of IoV technologies. A citation network was constructed to compile information regarding the development and evolution of IoV technologies up to the date that this paper was composed. Subsequently, the complex network structure of the citation network was digitized. MPA was performed, with an algorithm used to obtain the traversal counts of each link. Finally, key links with high traversal counts were connected to establish the main path of knowledge flow in each domain.

MPA was developed by Garfield et al. [42], who used it to investigate the development of research on DNA through analysis of the literature citation network for that field of study. Building on the work of Garfield et al., Hummon and Doreian [43] proposed a weighted calculation method, which laid the foundation for MPA. The weighted calculation method comprises two steps, namely information flow calculation and path tracing. Information flow is determined using algorithms that calculate the importance of citation relationships in the literature and assign weights to each connection in the network. The algorithms that are commonly used for weighted calculation include search path count (SPC), search path link count (SPLC), and search path node pair (SPNP). The main path is constructed by calculating path weights. SPC, the simplest algorithm, counts the number of times a particular link is traversed along all possible paths from all sources to all sinks. In SPLC, the number of times a link is traversed along all possible paths from all ancestors of a tail node (including itself) to all sinks is calculated. In SPNP, the number of times a link is traversed along all possible paths from all ancestors of a tail node (including itself) to all descendants of the head node is calculated [44].

On the basis of the suggestions of Liu and Lu [45], the present study combined global MPA and key-route MPA and empirically determined that SPLC outperformed SPC and SPNP and enhanced MPA. When weight is calculated in SPLC, all ancestor nodes of a tail node are considered to be starting points for each link in a network. The number of possible paths from the starting points that contain a given link and the number of possible paths from the descendants of a head node to all sink nodes are calculated. The product of the two numbers is the SPLC value for the link; a higher SPLC value indicates greater information flow through the link and that the link has a more pronounced effect on a field. SPLC is used to calculate the total number of possible paths from sources and nodes to tail nodes and the number of possible paths from tail nodes to sinks. The product of these numbers of possible paths then yields the weight for all links. For example, for the C–D link, the nodes A, B, and C occur before the tail node, and three potential paths include the link (i.e., A–C–D, B–C–D, and C–D). In addition, three paths from tail node D to sink nodes E, G, and H are possible (i.e., D–E, D–F–G, and D–F–H). Multiplying the numbers of potential paths (3 ancestor paths × 3 sink paths) yields 9 (Figure 1). Figure 2 is the results from the SPLC algorithm in determining the main path.

We used the MPA program developed by the Da Vincier Lab of National Taiwan University of Science and Technology to calculate main paths and cluster groups for the articles identified on the WOS database. We then used the MPA software (Release 465) and the Pajek program to evaluate the network links and visualize the network paths, with thicker paths indicating higher importance in the literature and the direction of the arrows indicating the direction of knowledge diffusion.

### 2.3. Edge-Betweenness Clustering

Newman and Girvan [46] employed cluster analysis to categorize similar studies within a citation network. When two articles cite the same study and are cited by the same group of studies, the two articles are likely to be discussing similar topics. Studies regarding the same topic form a group, or a “cluster”, within a citation network. The present study employed edge-betweenness clustering to analyze groups of studies by gradually removing important links within the citation network that we constructed. This process was completed using the following steps:We calculated edge-betweenness values, which represent the number of shortest paths between nodes within a network that include a given link and pass through a given edge.We removed the link with the highest edge-betweenness value in the citation network.We repeated Steps 1 and 2 if a cluster was separated from the citation network and then calculated the cluster’s modality. If no new clusters were separated, these steps were repeated until all links in the network were removed.We selected the clustering result with the highest modularity value. By completing edge-betweenness clustering, we obtained a sequence of link removal steps, with a cluster potentially being identified each time the steps were repeated. After the edge-betweenness clustering was completed, the cluster with the highest modularity value was selected.

## 3. Results

### 3.1. Data Statistics

We extracted academic studies from the WOS database. “Internet of Vehicles”, “Internet of Vehicle”, “vehicle to everything”, “V2X”, and other search terms related to the IoV were used to perform the search. An initial search of the WOS database yielded 8446 studies. After non-English articles were excluded, 8087 articles remained. After MPA was completed, 7519 studies remained, indicating that the constructed citation network had 93% precision (network size/number of papers = 7519/8087 = 0.93). The cumulative number of IoV papers on WOS database is presented in Figure 3. The number of studies available on the WOS database increased each year, indicating that the IoV is a trending topic that warrants further investigation.

#### 3.1.1. Influence of Journals

This study used the g-index as an indicator to rank the journals in which the identified articles were published on the basis of their influence in the field of the IoV. When the g-index values were the same for two journals, the h-index was used (Table 2). The highest ranked journal was IEEE Transactions on Vehicular Technology, which had a g-index score of 123 and had published 700 studies related to the IoV. The number of times this journal had been cited (24,131) was also high, indicating that the journal has had a strong influence in IoV research. The second-highest ranked journal, *IEEE Transactions on Intelligent Transportation Systems*, had a g-index of 107, had published 554 studies on the IoV, and had been cited 16,409 times. *IEEE Access* was ranked third, the *IEEE Journal on Selected Areas in Communications* was ranked fourth, and the *IEEE Communications Magazine* was ranked fifth.

#### 3.1.2. Influence of Authors

This study also used the g-index as an indicator to identify the 20 authors with articles listed on the WOS database with the strongest influence on research regarding the IoV. When two authors’ g-index values were the same, these authors’ h-index scores were used instead. The 20 most influential authors are listed in Table 3. Notably, because this ranking was established solely on the basis of the studies listed on the WOS database, influential authors in this field, including experts and editors who reviewed studies, that were not listed on the database may have been overlooked.

### 3.2. Global MPA

This study analyzed 22 academic articles through global MPA, as presented in Figure 4. In this figure, the green node represents the source node for research in the IoV field, and the blue node represents the sink node. Each red node represents a paper, and the nodes are connected by arrows indicating the flow of knowledge, with the thickness of the connecting links indicating weight and influence. The codes next to the nodes represent the first authors’ surnames, the first initials of the other authors’ surnames, and the publication years of the papers represented by the nodes. For example, for code GiYT2022, “Gi” is the first author’s surname; “Y” and “T” are the first letters of the second and third authors’ surnames, respectively; and 2022 is the year of publication.

The key research found through global MPA is as follows:Initial research: 2003 Medium Access Control (MAC)

MAC plays a critical role in wireless communication in the IoV and is primarily used to manage network access on communication interfaces, assign access rights for transmission media, and define rules for preventing conflict among devices sharing a single transmission medium. In the IoV, dedicated short-range communications (DSRC) technology specifically designed for vehicle communication can be used for wireless-communication-based traffic management to ensure road safety and prevent traffic congestion. The MAC layer of DSRC technology manages data transmissions and network access in V2V communication and ensures accurate and timely data transmission [47].

2.Phase I: V2V Communication (2006–2014)

In the first phase of IoV research, most studies explored V2V communication channels and wireless communication channels for data transmission between vehicles. These channels involve the network layer of the IoV architecture. When these channels experience interference, the quality and reliability of communication may be affected. The quality and reliability of V2V communication channels can also be affected by the distance between vehicles, the type of communication technology involved, the surrounding environment, obstacles, and reflective surfaces [48]. In addition, technology can be used to prevent interference in V2V communication channels. A key characteristic of V2V channels is their time variability and lack of stationarity, which can affect the reliability and latency of data packet transmission [49]. In summary, V2V communication channels play a critical role in facilitating communication between vehicles and are essential in ensuring that the IoV can support various applications and services.

3.Phase II: Device-to-Device (D2D) Management (2015–2017)

D2D vehicular communication in the IoV supports direct communication between devices without the need for signals to pass through a central infrastructure. D2D communication enables vehicles to exchange information and improves environmental sensing capability, traffic management, and safety. Key challenges associated with D2D communication include interference coordination, location-based resource allocation, and adjacent cooperative transmission scheduling [50]. The main advantages of D2D communication are higher spectrum efficiency, lower communication latency, lower energy consumption, and extended wireless coverage.

4.Phase III: Layer Perspective (2017–2018)

The two articles published during this phase of IoV research explored the physical and network layers of the IoV architecture. The physical layer of the architecture is responsible for transmitting and receiving signals through communication channels. One published study related to this layer focused on wireless communication between vehicles, the design and analysis of communication protocols, modulation and demodulation, encoding and decoding, multiuser detection, multiantenna technology, and beamforming [51]. The other article published during this phase investigated V2V and V2I communication at the network level, network topology, routing protocols, network layer communication protocols, data transmission, and congestion control [52].

5.Phase IV: Non-Orthogonal Multiple Access (NOMA) (2019–2021)

NOMA technology is an emerging technology used in IoV networks that helps devices connect to a network and share resources [53]. NOMA is used to assign a different power level to each user in order to enable sharing in situations where multiple users can use a single time or spectrum resource. This system improves bandwidth utilization and prevents interference. For IoV systems, NOMA offers several advantages. For example, it enables real-time traffic management, V2V communication, and remote vehicle control. In an IoV network with NOMA, vehicles can exchange information in real time by using a single communication resource, leading to more efficient traffic management and safer driving. NOMA can also be used for resource-intensive IoV applications and services and can improve ITSs [54].

6.Phase V: 6G Communication (2022)

Numerous studies have conducted conceptual research on 6G mobile wireless systems. IoV networks are highly dynamic and complex, and users of these networks expect them to have ultra-low latency, high reliability, and strong data connections, all of which can be achieved using 6G technology. In addition, the development of 6G technology could facilitate the creation of ITSs and overcome the limitations of 5G technology [55]; compared with 5G networks, 6G networks are expected to have higher data transmission rates, lower latency, and higher reliability and thus will likely be better able to support innovative IoV applications and services [56].

### 3.3. Key-Route Main Paths of Reviewed Articles

Because only a single main path is established through global MPA, gaining a comprehensive understanding of technological development through such analysis is difficult. Therefore, in the current study, we also used the key-route main path approach (Figure 5) to ensure that no influential studies would be overlooked. We identified 31 key studies regarding the key-route main path. Of these studies, 22 were also identified on the global main path. In the nine remaining studies, we discovered a topic that had not been researched in the aforementioned literature, namely software-defined networking in the IoV, which involves the use of software-based network management and control to separate the control layer from the data layer in an IoV network. Such separation enables dynamic and programmable network resource management and efficient and flexible communication between vehicles and infrastructural devices [57].

### 3.4. Cluster Analysis

To further explore research regarding the IoV, this study used the MPA program’s group finder function and edge-betweenness clustering analysis. We identified 20 clusters, and the top 7 clusters were selected for analysis. These groups, which were ordered on the basis of the year in which the first study in the group was published, comprised 335, 374, 2110, 1322, 620, 476, and 190 studies. Wordle was used to mine title and keyword data for each group. The following keywords were obtained for the seven clusters: 1. wireless channels, 2. networks and control, 3. VANETs, 4. security and privacy, 5. resource allocation and optimization, 6. traffic control, and 7. computing. We read the key studies in each group and assigned the following names to the seven cluster groups: 1. wireless channels, 2. communication protocols, 3. VANETs, 4. security and privacy, 5. resource allocation and optimization, 6. vehicle autonomous cruise control (ACC), and 7. deep learning and edge computing.

#### 3.4.1. Cluster 1: Wireless Channels (2002–2022)

Cluster 1 comprised 335 papers, including 18 key papers that were identified on the global main path (Figure 6); the main topics were RF channel emulation, wireless system designs based on IEEE 802.11p, non-stationary vehicular channels, V2V propagation channels, and channel nonstationarity and consistency.

Wireless channels in the IoV transmit signals; these signals can be affected by signal strength, multipath propagation, shadowing, and interference. The performance of wireless channels affects the quality and reliability of the IoV and can be influenced by the frequency bands that are used, the communication technology that is used, the communication environment, buildings, obstacles affecting the transmission range, and interference from devices and signals. Research into and development of reliable and efficient wireless channel technologies are crucial to the development of the IoV [58]. Wireless channels in the IoV entail rapidly changing propagation conditions because both the transmitters and receivers of such channels move and because the scattering environment changes quickly. One study reported that on European highways, V2V communication occurs at speeds as high as 400 km/h, which leads to high channel dynamics as vehicles drive past one another; in addition, advanced decision-directed channel estimation techniques are required in such situations [59]. In other complex and challenging environments, such as elevated bridges and tunnels, interference often occurs in wireless channels because of shadowing, scattering, and delay spreading [60]. Channel technology is directly related to information exchange and data transmission between facilities and is a vital aspect of the IoV and its development and application.

#### 3.4.2. Cluster 2: Communication Protocols (2003–2022)

Cluster 2 comprised 374 studies, including 16 key studies identified on the global main path (Figure 7). The main topics in the Cluster 2 studies were cooperative vehicle safety systems, access control protocols, DSRC, channel capacity optimization schemes for safety application, MAC protocols with dynamic interval schemes, and time-division multiple access (TDMA).

Numerous regions and institutions have developed communication protocols and control techniques for the IoV to enable it to support a wide range of applications and functions, including DSRC, wireless access in vehicular environments, long-term vehicle evolution, ITSs operating in the 5 GHz frequency band, and cellular V2X [61,62,63,64,65]. The IEEE established IEEE 802.11p to enable V2V and V2I communication and IEEE 1609 to define standards related to IoV communication and to ensure the security and reliability of V2V, V2I, and D2D communication [66,67]. Multichannel MAC protocols, which can be used for IoV wireless communication, support V2V and V2I communication and enable vehicles to communicate through multiple wireless channels. In addition, these protocols can be adjusted to suit different network environments in order to improve network performance and reliability [68]. The TDMA protocol enables users to share bandwidth resources on a single wireless channel. In this protocol, communication time is divided into time slots and allocated to different users. Each user can transmit data during their designated time slot without interfering with the transmissions of other users. Abbas et al. [69] proposed a protocol called PDMAC, which overcomes the limitations of the conventional TDMA protocol. Specifically, the PDMAC protocol prioritizes communication messages to enable efficient and reliable message propagation with little delay and high coverage.

#### 3.4.3. Cluster 3: VANETs (2004–2022)

Cluster 3 comprised 2110 studies, including 17 that were identified on the global main path (Figure 8). The main research topics in this cluster were routing and broadcast protocols, opportunistic data aggregation and forwarding, unified frameworks of clustering approaches, data dissemination schemes, unmanned aerial vehicle assistance, and machine learning algorithms in VANETs.

VANETs are vehicular networks that self-organize (i.e., organize without infrastructural support) through wireless communication and exchange and the sharing of information between vehicles [70]. VANETs are an essential part of the IoV and enable connection between vehicles and facilities. The IoV has a broad range of applications and involves numerous technologies, communication between vehicles, and the processing and analysis of infrastructure and traffic data [3]. Wisitpongphan et al. [71] explored solutions to the broadcast storm problem with respect to the self-organization of vehicular networks and introduced techniques to suppress the broadcast storm—including the imposition of node-based constraints and the distribution of control—and methods to combine these techniques to improve the performance of a network. Tonguz et al. [72] proposed a distributed vehicular broadcast protocol called DV-CAST, which improved safety and communication between vehicles through multihop broadcasting. The studies included in Cluster 3 of the present study discuss how the self-organization of vehicular networks can be improved through the utilization of self-learning techniques, which can increase the efficiency of data convergence and forwarding by enabling systems to learn parameters, such as routes and transmission power. Chatterjee et al. [73] explored machine-learning-based routing methods, such as reinforcement-learning-based routing and deep-learning-based routing, and their applications in the self-organization of vehicular networks; that study also investigated the advantages and disadvantages of routing protocols and provided directions for future research. Numerous studies have explored and developed control techniques to improve the reliability, security, and efficiency of VANETs; such research has promoted the development and application of the IoV.

#### 3.4.4. Cluster 4: Security and Privacy (2006–2022)

Cluster 4 comprised 1322 studies, including 20 key studies identified on the global main path (Figure 9). The topics of these studies included security and privacy in vehicular communications, privacy-preserving authentication schemes, certificateless aggregate signatures, security models and solutions, and the prevention of attacks.

Security and privacy protection protocols are a critical element of the IoV. In IoV communications between vehicles and infrastructure, data must remain secure, and user privacy must be protected. In addition, intrusions and attacks must be prevented. Several security and privacy protection protocols have been developed to ensure the credibility, security, and privacy of sensitive information [74,75,76,77]. Some studies have proposed cryptography-based authentication protocols to ensure the security of vehicular communications and to enhance resilience against attacks, such as those involving forgery [78,79]. One solution that has gained attention is pairing-free certificateless identity authentication schemes that support batch verification; such schemes ensure the efficiency of identity authentication and data privacy protection and improve a system’s scalability and security. This solution can be implemented to improve the applicability of the IoV [80,81].

#### 3.4.5. Cluster 5: Resource Allocation and Optimization (2006–2022)

Cluster 5 comprised 620 studies, including 16 key studies identified on the global main path (Figure 10). The topics of the studies in Cluster 5 included resource-sharing schemes, D2D resource allocation, a network perspective regarding radio resource allocation, resource allocation protocols for NOMA, and resource allocation and optimization for backscatter-enhanced NOMA.

Resource allocation and optimization in the IoV involves the efficient management of network resources, bandwidth, power, and computing resources to support the exchange of large amounts of communication data. The probability of outages should be minimized, and the outage capacity should be high to ensure that vehicular channels can optimally and instantaneously transmit information [82,83]. Studies have proposed frameworks for managing resource allocation, optimizing spectrum allocation, and preventing interruption and interference as a means of increasing the efficiency of the IoV [84,85]. In addition, numerous studies have noted that the application of NOMA technology in multiuser communication systems can facilitate the optimization of communication infrastructure. Finally, multiobjective optimization of channel quality, power, and reliability can improve a system’s overall performance, prevent communication delays, and improve the efficiency of communication and energy utilization [54,86,87].

#### 3.4.6. Cluster 6: Vehicle ACC (2010–2022)

Cluster 6 comprised 476 studies, including 14 key studies on the global main path (Figure 11). The topics of these studies included cooperative driving, acceleration-based connected cruise control, stability for large connected vehicle systems, connected and automated vehicles, and cooperative ACC (CACC).

ACC is an intelligent system in which sensors, control systems, and communication technologies are used to automate driving, increase comfort and safety, prevent accidents and traffic congestion, and save fuel and energy [88]. The CACC system enables communication and coordination between vehicles to improve driving safety and efficiency. Specifically, CACC reduces the complexity of a system and developmental costs and offers interoperability among types and models of vehicles. It also prevents traffic accidents, improves efficiency, and reduces fuel consumption [89,90]. Researchers have explored the effects of platoon-based cooperative driving in automated driving systems on traffic flow and stability, and studies have analyzed models of traffic dynamics and their relationships with IoV control systems to determine their effects on these systems’ performance and stability. By adopting the Lighthill–Whitham–Richards model, one study was able to facilitate interaction between traffic flow and IoV control systems; this interaction mitigated the effects of IoV control systems on traffic flow [91]. ACC and cooperative driving are crucial to the development of IoV technology because they can ensure vehicle safety and fuel efficiency by improving traffic flow while only minimally affecting the environment.

#### 3.4.7. Cluster 7: Deep Learning and Edge Computing (2013–2022)

Cluster 7 comprised 190 studies, including 10 key studies identified on the global main path (Figure 12), which covered topics such as data transmission, real-time interactive systems, deep learning in the edges of vehicles, mobile edge computing, and computation to offload task scheduling.

Deep learning and edge computing are two key technologies used in the IoV. Deep learning algorithms can analyze large amounts of data obtained from sensors and other sources to improve the efficiency of vehicle communication and data exchange. Edge computing involves processing data and computations at the network edge, which is close to the data source, to reduce signal latency, loosen bandwidth requirements, and improve the performance and efficiency of vehicle communications and data exchange [92]. The earliest study in Cluster 7, which was published in 2013, investigated data tracking and data transformation problems in the IoV; that study proposed two solutions, namely area-based tracking and parked-vehicle-assisted tracking [93]. Compressive sensing (CS) is a highly efficient sampling method that is more efficient than the conventional Nyquist sampling theorem. CS has been used to monitor driving safety, ensure privacy, detect vehicles, facilitate communication, evaluate traffic conditions, stream videos, and recognize objects [94]. Deep reinforcement learning is a machine learning method based on deep learning and has various applications in the IoV. It can perform complex tasks, such as predicting traffic flow, planning routes, and guiding autonomous driving [95]. Sensor data needs to be rapidly processed in the IoV, and thus deep learning and edge computing are crucial to the development of IoV technology.

## 4. Discussion

### 4.1. Research Regarding the IoV

We used the logistic growth model and Loglet Lab 4 to graphically present the research regarding the IoV. In Figure 13, the dashed lines represent the estimated cumulative number of papers, and the solid lines and dots represent the actual cumulative number of papers. In addition, the period, limit, and inflection point were estimated using the algorithm.

The number of IoV-related papers has increased considerably in recent years; since 2018, the number of IoV-related papers published per year has increased by more than 1000. According to the results of the Loglet Lab 4 analysis, the inflection point is forecasted to occur in 2025, and research regarding the IoV is currently, at the time of writing, in a period of rapid growth. The number of studies regarding the IoV will then likely stabilize in 2038, during which the maximum number of studies likely to be conducted is 18,000.

### 4.2. Key Topics and Trajectory of Research Regarding the IoV

In the present study, global MPA revealed that future research regarding the IoV is likely to cover topics related to all four layers of the IoV architecture, including the topics of MAC, V2V channels, D2D communications, layers, NOMA, and 6G communications. Table 4 presents the cluster analysis results, including the main keywords of the seven clusters and their frequencies, word clouds of the keywords, and the trajectory of research for each cluster. The topics investigated in the most studies were security and privacy protection protocols, VANETs, and ACC.

### 4.3. Academic Development and Emerging Topics Related to the IoV

Emerging topics were identified using text mining and pivot analysis. We divided the 7519 studies into two groups: one with 3988 studies published between 1993 and 2019 and one with 3531 studies published between 2020 and 2022. Subsequently, we used text mining and pivot analysis to identify high-frequency keywords in these groups. In addition, we performed text mining on the titles and abstracts of the studies to identify the 200 most frequently occurring keywords. Finally, we conducted a differential analysis to compare the datasets and identified the following emerging topics: blockchain, deep learning, edge computing, cloud computing, vehicle dynamics, and 6G mobile communications (Table 5).

Blockchain:

Blockchain technology can be used in the IoV to achieve the decentralization and distribution of data in order to ensure the security and reliability of vehicle communication and data exchange. Blockchain-based IoV communication systems are secure and reliable and can solve problems related to trust, authentication, authorization, data sharing, privacy protection, smart contracts, and decentralization in network communication systems [96]. The advanced encryption techniques of blockchain technology can ensure the authenticity and integrity of data.

2.Deep learning:

Deep learning involves the use of advanced machine learning algorithms and neural networks to process and analyze large amounts of data generated through IoV ecosystems and has been used for vehicle identification, vehicle behavior analysis, vehicle communication, vehicle safety, real-time traffic management, collision avoidance, and autonomous driving. Deep learning algorithms can ensure the security and efficiency of communication and data exchange in IoV networks [97,98].

Cooperative perception has emerged as a means through which vehicle networks can adapt to ever-changing traffic environments. This technology involves the sharing and exchange of information between vehicles, such as those that are autonomous and those with perception capabilities, to coordinate and expand their perceptions of the surrounding environment, which can lead to improved road safety and traffic efficiency. Deep learning is vital for object detection and identification, data fusion, and the development of communication and collaboration strategies required for cooperative perception. In autonomous vehicles, deep learning, which uses perception data collected from vehicle perception systems (e.g., radars, LiDAR, and cameras), can be applied in three-dimensional (3D) LiDAR systems to perform data acquisition and analysis and in Hydro-3D systems to perform object detection and identification. When an individual is driving in a complex traffic environment, their ability to accurately distinguish between objects (e.g., pedestrians, vehicles, and traffic signs) is vital to ensuring that the vehicle is safely operated. Deep learning can also be applied for data fusion, which involves data from multiple sources (e.g., different vehicles or perception systems) being combined to obtain a more comprehensive perception of the environment. Finally, deep learning algorithms can be applied in communication and collaboration strategies focused on optimizing the transmission and sharing of perception data between vehicles to determine when data should be shared as well as how data received from other vehicles should be processed. These applications of deep learning can improve the efficacy and efficiency of cooperative perception [99].

3.Edge computing:

Edge computing involves data processing and computation at the edge of a network, which is near the data source. Edge computing can be implemented to reduce the latency and bandwidth requirements of IoV applications and thus can improve the performance and efficiency of automotive communication and data exchange. In addition, edge computing eliminates the need to transmit all data and computations to centralized data centers for processing; instead, processing is performed in vehicles or on other devices. Integrating edge computing into the IoV can lead to improved communication and enhanced data exchange performance and efficiency [100,101].

4.Cloud computing:

Cloud computing involves supporting data exchange and processing through cloud services. In-vehicle devices can transmit data to the cloud for processing and analysis, and subsequently, these data are transmitted back to in-vehicle devices or to other devices or facilities. Cloud computing can reduce computational and storage requirements for in-vehicle devices to save on cost and reduce energy consumption [102,103].

5.Vehicle dynamics:

Vehicle dynamics involve a vehicle’s motion and behavior in dynamic environments and involve V2V communication, cooperative control, perception-based vehicle control, prediction-based vehicle control, and vehicle control in ITSs. Vehicle dynamic models can be used to analyze the motion, acceleration, and braking of vehicles under various driving conditions and various environmental factors. This information can then be used to develop and optimize IoV applications [104].

Xia et al. proposed an algorithm based on consensus and vehicle kinematic or dynamic synthesis for sideslip angle estimation and adopted a multisensory framework to increase the reliability and accuracy of their estimation. They described the importance of the sideslip angle of vehicles and discussed its role in vehicle stability, trajectory planning, road condition estimation, and mode switching in autonomous driving systems. In addition, they used a global navigation satellite system (GNSS) for vehicle sideslip angle estimation. However, the GNSS had poor estimation accuracy, particularly when horizontal speed was being measured, and it also had poor reception in tunnels and urban canyons. Therefore, the applicability of the active safety system was limited. Liu et al. developed a novel approach to vehicle slip angle estimation, which is crucial to achieving stability and control in automated vehicles; that study designed a vehicle attitude angle observer and estimation approach and explored the possibilities of using multiple sensors (e.g., a GNSS and an inertial measurement unit) for data collection and fusion to enhance the accuracy and efficiency of VSA estimation. Additionally, they applied the method to an actual driving scenario to verify its effectiveness [105,106].

6.Application of 5G and 6G technologies:

Application of 5G and 6G communication technologies can increase the bandwidth and speed, decrease the latency, increase the reliability, ensure the security and privacy, and improve the encryption and authentication of the IoV. In addition, the development of 5G and 6G communication technologies can lead to more efficient, reliable, and secure communication in the IoV, which in turn can facilitate the development of ITSs and enhance their applicability [107].

## 5. Conclusions

The development of innovative IoV technologies has considerably affected the automobile industry. Therefore, developing an in-depth understanding of novel IoV technologies and their potential applications in the automobile industry can enable researchers and technicians to improve vehicle driving safety, elevate the overall performance of the automobile industry, and overcome future challenges. Although numerous studies have investigated applications of IoV technologies in the automobile industry, to the best of the present authors’ knowledge, no study had yet employed MPA to investigate this topic. Thus, the current study used MPA to determine the main knowledge development trajectories of IoV research. This study reviewed 7521 papers published between 1993 and 2023. The results of the present MPA can provide researchers and technicians with an overview of the research on the IoV and applications of IoV technologies in the automobile industry. The present findings reveal that the knowledge development trajectories in IoV research can be divided into five phases. In Phase I (2006–2014), studies mainly explored V2V communication channels and wireless transmission channels for data transmission in the IoV architecture, and these studies primarily focused on the network layer of the IoV architecture. In Phase II (2015–2017), studies examined applications of D2D communication technologies in the IoV, which enable direct communication between devices and do not require a central infrastructure. In Phase III (2017–2018), two papers delved into the physical and network layers of the IoV architecture. In Phase IV (2019–2021), studies mainly investigated NOMA, a technology used in IoV networks that enables devices to simultaneously connect to networks and share resources. Finally, the studies in Phase V (2022) have indicated that 6G networks are likely to lead to higher data transmission rates, lower latency, and higher reliability and can therefore support the development of innovative IoV applications and services.

The present study conducted cluster analysis and identified seven main cluster groups in the research regarding the IoV, namely wireless channels, networks and control, VANETs, security and privacy, resource allocation and optimization, traffic control, and computing. The results revealed that most related studies have focused on technology optimization, information security, resource allocation, and coordination and collaboration. Additionally, analyses of influential authors, influential journals, and the growth curve model of research regarding the IoV were presented using descriptive statistics to reveal the distribution of published studies and the growth trend of studies regarding the application of IoV technology in the automobile industry. According to our results, the inflection point of the literature growth curve will occur in 2025. At the time of writing, research regarding the IoV is in a period of rapid growth. The number of studies being published on the topic of the IoV is then expected to stabilize in 2038, during which the maximum number of studies that is likely to be published is 18,000. Finally, research regarding the IoV is expected to enter the maturity phase in 2030 and reach saturation in 2035.

This study conducted data mining and pivot analysis to identify emerging topics in IoV research. The application of blockchain technology in the IoV can ensure security in vehicle communication and data exchange. In addition, the decentralized database systems on the blockchain can ensure the safe and reliable transmission and storage of data and can therefore lower the risk of malicious attacks. Deep learning has been extensively applied in autonomous driving technology connected to the IoV and can be used to teach autonomous vehicles how to learn and perceive complex traffic conditions and ensure that these vehicles can quickly and accurately respond to traffic conditions. The development of edge computing and cloud computing technologies has also led to greater innovation in IoV research. In edge computing, a data processing task is performed near the data source (i.e., the vehicle) to considerably reduce communication latency and improve the real-time response capability. Cloud computing, which has considerable storage and processing capabilities, enables vehicles to share data and learning outcomes and thus accelerates the overall learning and adjustment speed of the IoV architecture. Research regarding vehicle dynamics has enabled researchers and technicians to accurately predict vehicle behaviors, and this improved accuracy has increased traffic system management efficiency. Advancements in 5G and 6G mobile communication technologies are expected to elevate V2V and V2I transmission speeds and create new opportunities for the development of the IoV. The results of research regarding these emerging topics are likely to enable the identification of new possibilities for the future development of the IoV.

## Figures and Tables

**Figure 1 sensors-23-06120-f001:**
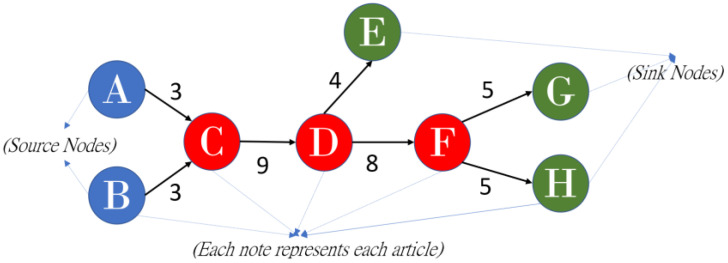
Example of the SPLC algorithm.

**Figure 2 sensors-23-06120-f002:**
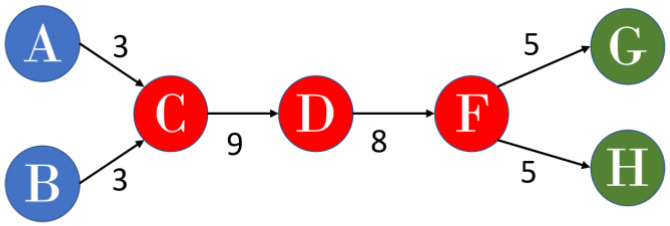
The results of SPLC algorithm final main path.

**Figure 3 sensors-23-06120-f003:**
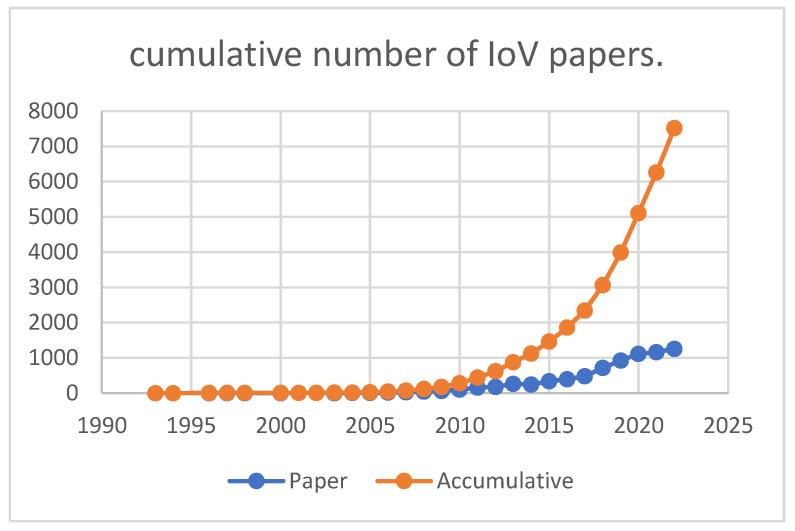
Cumulative number of IoV papers on WOS database.

**Figure 4 sensors-23-06120-f004:**
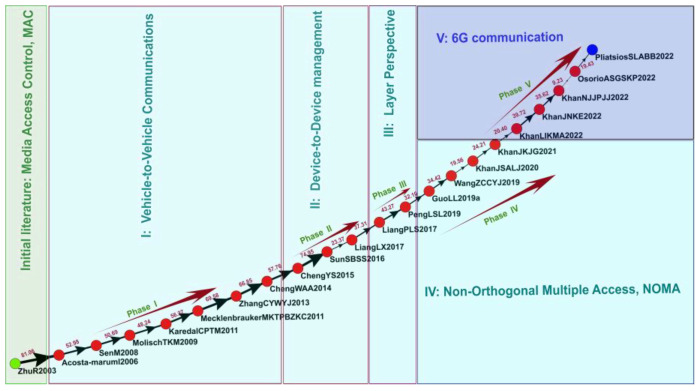
Global main path.

**Figure 5 sensors-23-06120-f005:**
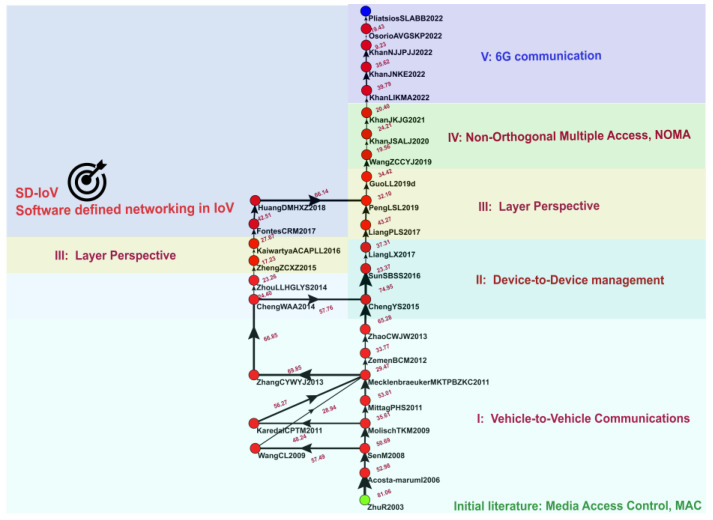
Key-route main path.

**Figure 6 sensors-23-06120-f006:**
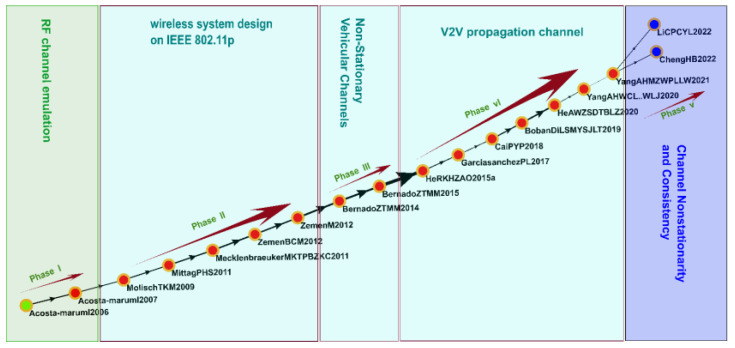
Analysis of Cluster 1: wireless channels.

**Figure 7 sensors-23-06120-f007:**
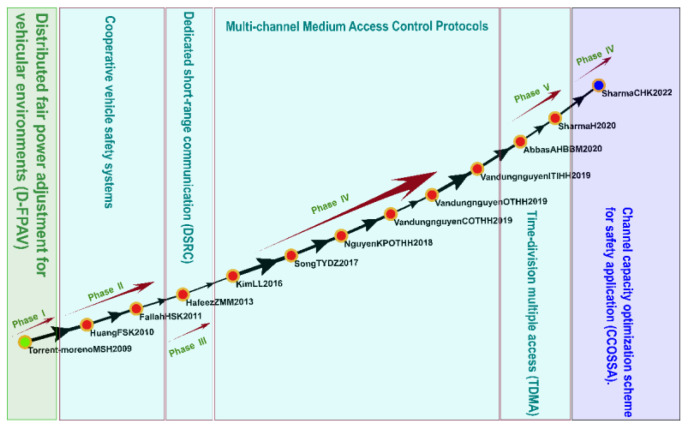
Analysis of Cluster 2: communication protocols and control techniques.

**Figure 8 sensors-23-06120-f008:**
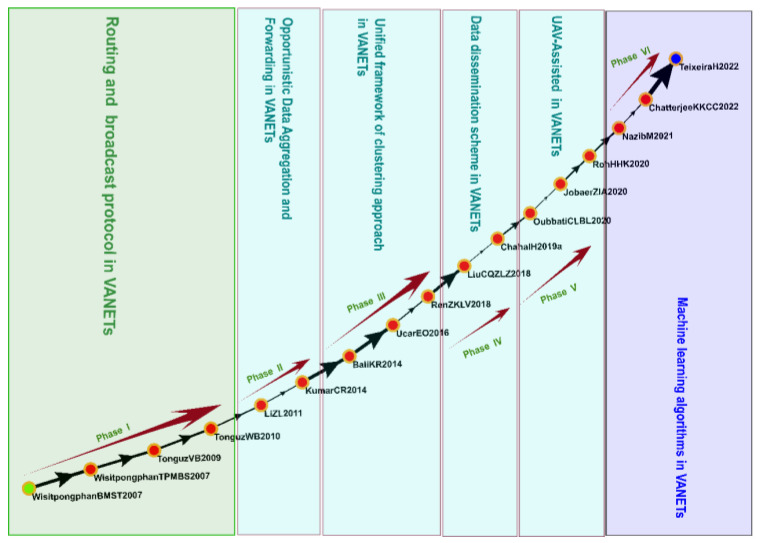
Analysis of Cluster 3: VANETs.

**Figure 9 sensors-23-06120-f009:**
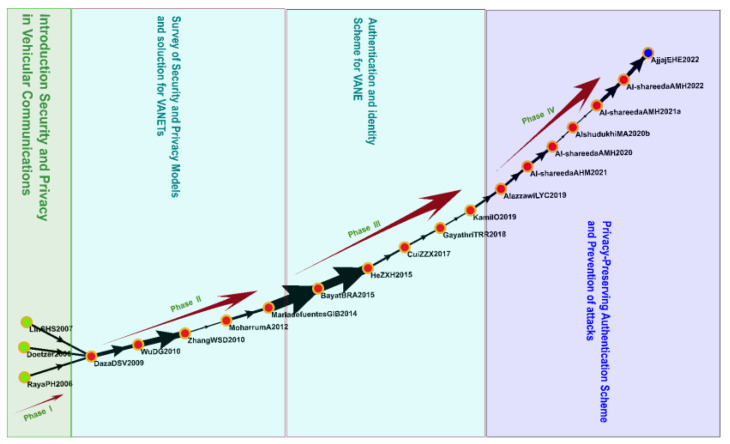
Analysis of Cluster 4: security and privacy protection protocols.

**Figure 10 sensors-23-06120-f010:**
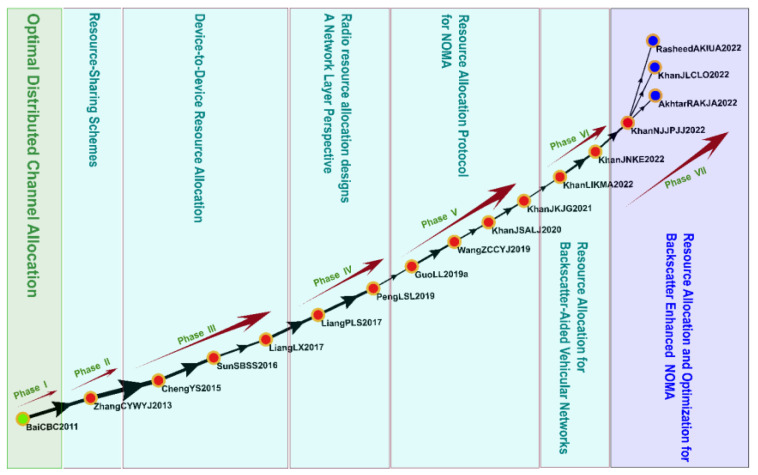
Analysis of Cluster 5: resource allocation and optimization.

**Figure 11 sensors-23-06120-f011:**
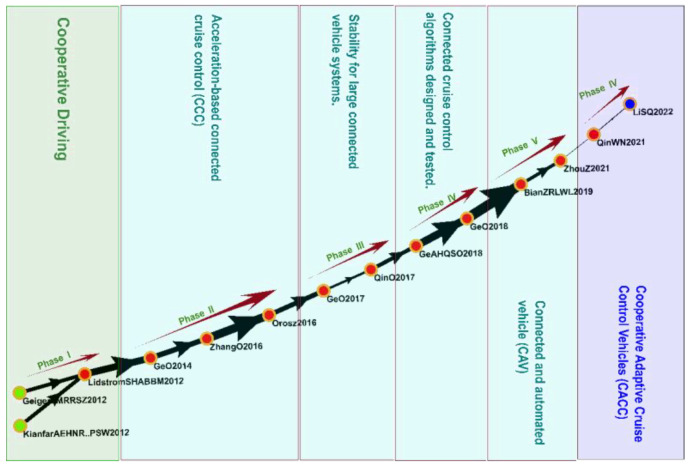
Analysis of Cluster 6: vehicle ACC.

**Figure 12 sensors-23-06120-f012:**
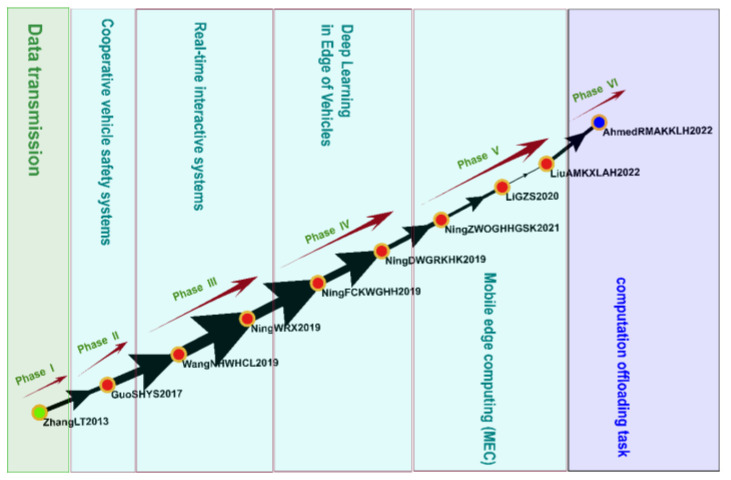
Analysis of Cluster 7: deep learning and edge computing.

**Figure 13 sensors-23-06120-f013:**
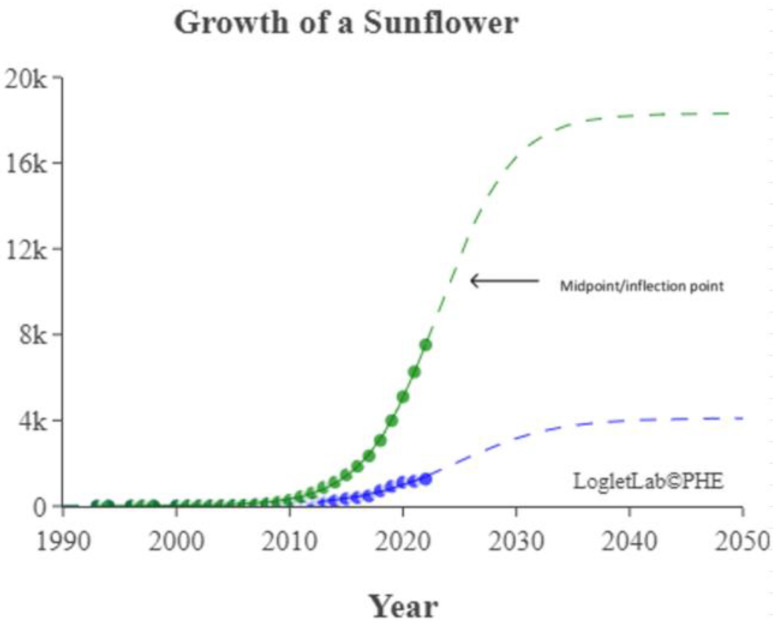
Trajectory of research regarding the IoV.

**Table 1 sensors-23-06120-t001:** Summary of the literature regarding IoV architectures (adapted from [31]).

Author	Layers	Year	Name of Layers	Communication Links Models
Nanjie [13]	3	2011	(1) Client, (2) Connection, (3) Cloud	V2V, V2R, V2P, V2I
Li et al. [14]	4	2012	(1) Sensing, (2) Network, (3) Data, (4) Application	V2V, V2I
Sherazi et al. [15]	3	2019	(1) Client, (2) Connection, (3) Cloud	V2I
Bonomi [16]	4	2014	(1) Services, (2) Operation, (3) Infrastructure, (4) End points	V2V, V2I
He et al. [17]	4	2016	(1) Cloud computing, (2) SDN control (3) Fog computing, (4) Infrastructure	V2V, V2I
Kaiwartya et al. [1]	5	2016	(1) Perception, (2) Coordination, (3) Artificial intelligence, (4) Application, (5) Business	V2I, V2V, V2S, V2P, V2R
Contreras-Castillo et al. [18]	7	2016	(1) User interaction layer, (2) Data acquisition layer, (3) Data filtering and pre-processing layer, (4) Communication layer, (5) Control and management layer, (6) Business layer, (7) Security layer	V2S, V2P, V2V, V2I, V2R, R2R
Wang et al. [19]	3	2017	(1) Physical, (2) Control, (3) Application	V2I, V2V, V2S, V2P, V2R
Wan et al. [20]	3	2017	(1) Vehicle, (2) Location, (3) Cloud	V2V, V2R
Gandotra et al. [21]	3	2017	(1) D2D area network, (2) Network management, (3) D2D applications	(D2D-B), (D2D-C), direct D2D (D2D-D), (D2D-N).
Yang et al. [2]	4	2017	(1) Vehicle network environment sensing and control (2) Network access and transport, (3) Coordination computing control, (4) Application	V2V, V2I
Nahri et al. [22]	3	2018	(1) IoV, (2) Fog, (3) Cloud	V2I, V2V
Contreras-Castillo et al. [9]	7	2018	(1) Vehicle interface, (2) Data acquisition, (3) Data filtering and preprocessing, (4) Communication, (5) Control and management, (6) Processing, (7) Security	V2V, V2I, V2P, V2S, V2R, V2D
Li-minn et al. [5]	7	2018	(1) Identification, (2) Physical objects, (3) Inter-intra devices, (4) Communication, (5) Cloud services, (6) Multimedia and big data computation, (7) Application	V2V, V2R, V2X, V2G, V2S, V2I, V2B, V2H, V2P, V2D, D2D
Kai Liu et al. [23]	4	2019	(1) Cloud computing, (2) SDN control, (3) Fog computing, (4) Infrastructure	I2V, V2V
Ji et al. [24]	4	2020	(1) Cloud platform, (2) Edge, (3) Data acquisition, (4) Security authentication	V2V, V2I, V2R, V2P
Li et al. [25]	5	2020	(1) Physical, (2) Data link, (3) Network, (4) Perception, (5) Application	V2V, V2R
ICAISC-2020 [26]	5	2020	(1) Business, (2) Application, (3) Artificial intelligence, (4) Coordination, (5) Perception	V2R, V2I, V2X
Nassar and Yilmaz [27]	3	2021	(1) Deep reinforcement learning, (2) Infrastructure network, (3) Management	V2X, V2I, V2R
Gao et al. [28]	3	2022	(1) Clients, (2) Endorsement and commitment peers, (3) Ordering services	V2X, V2R
Wang et al. [29]	4	2023	(1) Terminal, (2) Network, (3) Platform, (4) Service	V2X
Mao et al. [30]	4	2023	(1) Data plane, (2) Lower control plane, (3) Upper control plane, (4) Application plane	V2R, V2I, V2X

**Table 2 sensors-23-06120-t002:** Top 10 most influential journals in IoV field.

Ranking	Name	g-Index	h-Index	Total Papers	Active Years
1	IEEE Transactions on Vehicular Technology	123	83	700	1997–2022
2	IEEE Transactions on Intelligent Transportation Systems	107	70	554	2006–2022
3	IEEE Access	75	45	574	2015–2022
4	IEEE Journal on Selected Areas in Communications	69	40	72	2007–2022
5	IEEE Communications Magazine	68	37	68	2003–2022
6	IEEE Internet of Things Journal	66	37	244	2014–2022
7	Transportation Research Part C: Emerging Technologies	62	36	107	2005–2022
8	Computer Communications	52	29	98	2007–2022
9	Ad Hoc Networks	49	32	119	2009–2022
10	Vehicular Communications	47	29	175	2014–2022

**Table 3 sensors-23-06120-t003:** Top 10 most influential authors in IoV field.

Ranking	Name	g-Index	h-Index	Total Papers	Active Years
1	Rodrigues, Joel J. P. C.	41	23	45	2011~2022
2	Guizani, Mohsen	39	25	52	2014~2022
3	Shen, Xuemin (Sherman)	37	27	37	2008~2018
4	Ai, Bo	37	18	41	2013~2022
5	Kumar, Neeraj	36	21	52	2013~2022
6	Chen, Chen	31	19	43	2012~2022
7	Calafate, Carlos T.	30	19	30	2010~2021
8	Zhong, Zhangdui	30	16	35	2013~2022
9	Shen, Xuemin	29	23	29	2007~2022
10	Boukerche, Azzedine	29	15	39	2008~2022
11	Gerla, Mario	28	20	28	2009~2019
12	Cano, Juan-Carlos	28	18	28	2010~2020
13	Vinel, Alexey	28	16	28	2011~2022
14	Zeadally, Sherali	27	18	27	2008~2022
15	Manzoni, Pietro	27	17	27	2010~2020
16	Wu, Celimuge	27	17	28	2010~2022
17	Wang, Cheng-Xiang	26	19	26	2009~2022
18	Cheng, Xiang	26	14	26	2009~2022
19	Choo, Kim-Kwang Raymond	25	14	26	2016~2022
20	Zhang, Yan	24	17	24	2010~2022

**Table 4 sensors-23-06120-t004:** Topics, number of papers, keywords, word clouds, and curves for each cluster.

Research Topic	Keywords	Word Cloud	Article Growth Trend
Cluster 1 Wireless Channels (2002–2022)	Channels (0.707) Vehicle-to-Vehicle (0.352) Model (0.146) MIMO (0.140) Modeling (0.098) Characterization (0.0746)	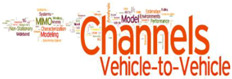	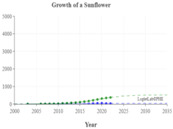
Cluster 2 Communication protocols (2003–2022)	MAC (0.280) Networks (0.272) Protocol (0.216) VANETs (0.171) Safety (0.168) Control (0.144)	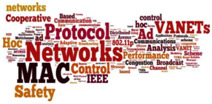	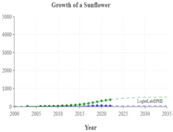
Cluster 3 VANETs (2004–2022)	VANETs (0.262) Networks (0.231) Ad (0.227) hoc (0.227) networks (0.171)	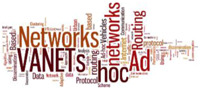	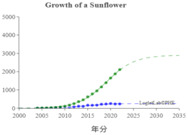
Cluster 4 Security and Privacy Protection Protocols (2006–2022)	Security (0.292) Privacy (0.265) Authentication (0.262) Networks (0.181) VANETs (0.180)	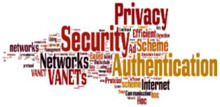	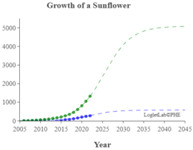
Cluster 5 Resource allocation and optimization (2006–2022)	Communications (0.475) Networks (0.317) V2X (0.214) Resource (0.161) Allocation (0.148) 5G (0.108)	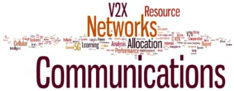	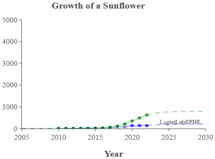
Cluster 6 Autonomous Cruise Control (ACC) (1994–2022)	Control (0.397) Connected (0.372) Traffic (0.223) Cooperative (0.135) Automated (0.098) Communication (0.098)	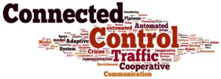	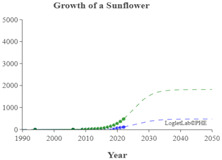
Cluster 7 Deep Learning and Edge Computing (2013–2022)	Computing (0.468) Internet (0.421) Edge (0.284) Offloading (0.273) Learning (0.200) Reinforcement (0.126) Deep (0.121)	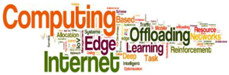	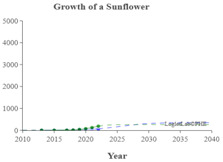

**Table 5 sensors-23-06120-t005:** Emerging topics in the field of IoV.

No.	Keywords	Cluster 1 Keyword Counting (1993–2019)	Cluster 2 Keyword Counting (2020–2022)	Emerging Topic
1	BLOCKCHAINS	32	261	◎
2	ROADS	8	225	◎
3	VEHICLE DYNAMICS	9	194	◎
4	5G MOBILE COMMUNICATION	39	190	◎
5	SAFETY	29	187	◎
6	DEEP LEARNING	19	181	◎
7	EDGE COMPUTING	23	155	◎
8	COMPUTATIONAL MODELING	4	123	◎
9	SENSORS	8	122	◎
10	CLOUD COMPUTING	29	112	◎
11	REINFORCEMENT LEARNING	11	79	◎
12	COMPUTER ARCHITECTURE	2	76	◎
13	REAL-TIME SYSTEMS	4	71	◎
14	6G MOBILE COMMUNICATION	0	58	◎

Differential analysis of keywords used in studies published between 2013 and 2019 and keywords used in studies published between 2020 and 2022. ◎ indicates that a keyword is related to an emerging topic (Cluster 1 < 50 and Cluster 2 minus Cluster 1 > 50).

## Data Availability

The data presented in this study are available on request from the corresponding author.

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
