# Peer review of "Knowledge Development Trajectory of the Internet of Vehicles Domain Based on Main Path Analysis"

_sensors, 2023, doi:10.3390/s23136120_

Round 1

Reviewer 1 Report

This study performed main path analysis (MPA) to investigate the trajectory of the research on the IoV based on data retrieved from the Web of Science (WOS) academic data base, identified emerging research topics by using data mining and statistical analysis, and offered several predictions regarding the research on the IoV on the basis of the results.

There are two shortcomings which seriously affect the publication of this article.

The sample size is too small

Just 22 articles were used for the Main Path Analysis, which is far from sufficient to support the study, makes the results unconvincing and further affects the predictions based on the results.

2Lack of innovation

There is no innovation in analytical methods based on data characteristics and application-specific need. In this case, adequate workload and in-depth analysis are indispensable. However, this article did not do it. The content of Table 1 should be supplemented by articles from 21-23 years, not carried directly from [13]. Besides, there are many simple statistical analysis and descriptive analysis, and in-depth analysis is lacking.

In summary, it is recommended to major revise this article.

Reviewer 2 Report

Based on the information from Web of Science database, and citation networks among the studies, a comprehensive study about main path analysis (MPA) to investigate the trajectory of the research on the Internet of Vehicles (IoV). MPA revealed that the research mainly covers media access control, vehicle-to-vehicle channels, device-to-device communications, layers, nonorthogonal multiple access, and sixth-generation (6G) communications. The trends of the IoV development has been identified: the main research topics related to the IoV include wireless channels, communication protocols, vehicular ad hoc networks, security and privacy, resource allocation and optimization, autonomous cruise control, deep learning, and edge computing. With these merits of the paper, I will recommend a minor revision of the work. Below are the comments for revisions:

- Please highlight the motivations and the contributions in the introduction.

- In section 4.3 (2), it will be great that the authors can mention the cooperative perception in the deep learning part as the deep learning has been widely used in cooperative perception for internet of vehicles. Also. I hope the authors can include some most recent literature about cooperative perception to discuss the works using deep learning: an automated driving systems data acquisition and analytics platform; hydro-3d: hybrid object detection and tracking for cooperative perception using 3D lidar.

- Please elaborate more on the caption of the figure 1. For the example, SPLC has not been defined yet. And what does the node in the figure mean?

- For section 4.3, in particular the vehicle dynamics section, it will be interesting if the authors can also discuss about a little about the state estimation problems about vehicle dynamics control by considering the literature in: autonomous vehicle kinematics and dynamics synthesis for sideslip angle estimation based on consensus kalman filter; automated vehicle sideslip angle estimation considering signal measurement characteristic.

- The quality of the figures can be improved. For instance, the context in Figures 3-9 are not very clear and can be improved.

- Please make the conclusions more concise so the readers can clearly identify the main conclusions that have been formed. 

Reviewer 3 Report

This paper proposed “Knowledge Development Trajectories of the Internet of Vehi- 2 cle Domain Based on Main Path Analysis”. I recommend following corrections.

1-     The introduction section needs more investigation of some recent and relevant work that has been done in the past. Also add the role of IoT in Internet of Vehicles. I suggest a few papers for your reference, “A Step toward Next-Generation Advancements in the Internet of Things Technologies, An Overview of Medium Access Control and Radio Duty Cycling Protocols for Internet of Things, Lidar Point Cloud Compression, Processing and Learning for Autonomous Driving”

2-     Carefully recheck table1, tabele2 and table5 configuring and table caption.

3-     Add more details of your proposed method.

4-     Table 5, last column symbol what stand for, you should explain in table caption.

5-     Page 18, line 533, deep learning font should be correct.

6-     Page 1, line 37, “intelligent transportation system” first character should small letter,

7-     Section 1 and Section 2 have a lot of unnecessary headings; it's better you write in the form of a paragraph.

8-     In the manuscript, there are many grammatical errors, formatting and typos. Carefully revised all manuscripts and corrected them.

Extensive editing of English language required

Round 2

Reviewer 1 Report

The article meets the requirements of publication and can be accepted.